# Local Masked Reconstruction for Efficient Self-Supervised Learning on High-resolution Images

## Abstract

Self-supervised learning for computer vision has achieved tremendous progress and improved many downstream vision tasks, such as image classification, semantic segmentation, and object detection. Among these, generative self-supervised vision learning approaches such as MAE and BEiT show promising performance. However, their global reconstruction mechanism is computationally demanding, especially for high-resolution images. The computational cost will extensively increase when it is scaled to a large-scale dataset. To address this issue, we propose local masked reconstruction (LoMaR), a simple yet effective approach that reconstructs image patches from small neighboring regions. The strategy can be easily integrated into any generative self-supervised learning techniques and improves the trade-off between efficiency and accuracy compared to reconstruction over the entire image. LoMaR is 2.5× faster than MAE and 5.0× faster than BEiT on 384×384 ImageNet pretraining, and surpasses them by 0.2% and 0.8% in accuracy, respectively. It is 2.1× faster than MAE on iNaturalist pretraining and gains 0.2% in accuracy. On MS COCO, LoMaR outperforms MAE by 0.5 $AP^{box}$ on object detection and 0.5 $AP^{mask}$ on instance segmentation. It also outperforms MAE by 0.2% on semantic segmentation. Our code will be made publicly available.

## 1 Introduction

Recently, self-supervised learning Chen et al. (2021); Caron et al. (2021); Bao et al. (2022); He et al. (2021); Chen et al. (2020a); Bachman et al. (2019); Wu et al. (2018); Oord et al. (2018); Hjelm et al. (2018) has achieved enormous success in learning representations conducive to downstream applications, such as image classification and object detection. Among these, several generative methods such as Masked Autoencoder (MAE) He et al. (2021) and Bidirectional Encoder Representation from Image Transformers (BEiT) Bao et al. (2022), which reconstruct the input image from a small portion of image patches, have demonstrated excellent performance.

However, a major bottleneck of MAE and BEiT is their high demand for compute, as they reconstruct masked image patches from global information and operate on a large number of image patches. For example, pretraining an MAE-Huge network on ImageNet under $224 \times 224$ resolution takes 34.5 hours on 128 TPU-v3 GPUs. BEiTBao et al. (2022) training is even slower due to the cost of the discrete variational autoencoder.

High-resolution images further exacerbate this issue, due to the $\mathcal{O}(n^2)$ time complexity of the Transformer model on $n$ image patches. For example, pretraining MAE on 384×384 images consumes 4.7 times the compute time of 224×224 counterpart. However, high-resolution images are essential in many tasks, such as object detection. Thus, improving the efficiency of pretraining holds the promise to unleash additional performance gains under pretraining with a much larger dataset or higher-resolution images.

We observe that most reconstruction in MAE relies on local information only. In Fig. 1, we visualize the attention weights (white indicating high attention) when reconstructing a target image patch . From a pretrained $MAE_{Large}$ model, we extract the attention weights from the decoder layers 2, 4, 6, and 8. The model mostly attends to patches close to the target patch, which motivates us to limit the range of attention used in the reconstruction.

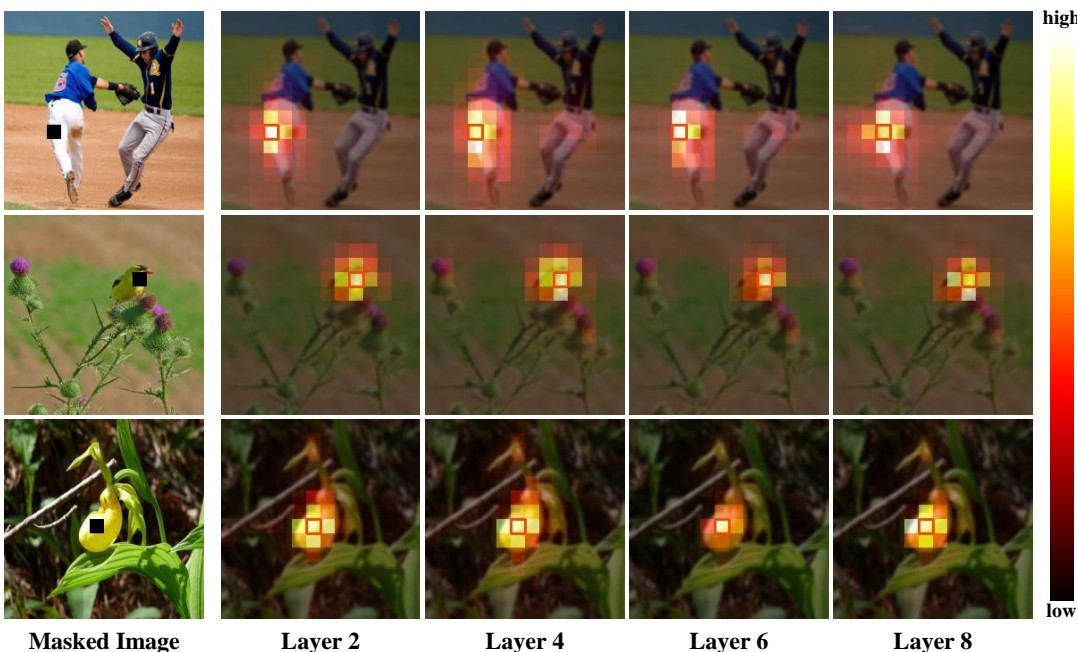

Figure 1: We visualize the attention patterns employed by $MAE_{Large}$ He et al. (2021) in the reconstruction of a random target patch, indicated by orange. Patches that are important for prediction are usually close to the target patch. We selected the images randomly from the ImageNet-1K Deng et al. (2009) Val set.

Hence we propose a new model, dubbed **Lo**cal **Ma**sked **R**econstruction or LoMaR. The model restricts the attention region to a small window, such as 7×7 image patches, which is sufficient for reconstruction. Similar approaches Child et al. (2019); Sukhbaatar et al. (2019); Zaheer et al. (2020) have been seen in NLP problems that need to process long sequences. The small windows have also been explored in vision domains for higher training and inference speed Liu et al. (2021); Yang et al. (2021). But unlike prior work in vision transformers, which create the shifting windows with fixed coordinates for each image. We instead sample several windows with random locations, which can better capture the objects in different spatial areas.

In Figure 2, we compare LoMaR and MAE and note two major differences: a) We sample a region with k×k patches to perform masked reconstruction instead of from the full number of patches. Instead of reconstructing the masked patches from the 25% visible patches globally located in the image, we find that it is sufficient to recover the missing information with only some local visual clues. b) We replace the heavy-weight decoder in MAE with a lightweight MLP head. We feed all image patches directly into the encoder, including masked and visible patches. In comparison, in MAE, only the visible patches are fed to the encoder. Experiments show that these architectural changes bring more performance gain to the local masked reconstruction in small regions.

After conducting extensive experiments, we found that

- LoMaR gains a large efficiency gain than other baselines in pretraining high-resolution images since its computation cost is invariant to the different image resolutions. However, other approaches have computational cost quadratic to the image resolution increase, which leads to much expensive pretraining. e.g., for pretraining on 448×448 images, LoMaR is 3.1× faster than MAE and 5.3 × faster than BEiT while achieving higher classification performance.

- LoMaR also has a strong generalization ability on other tasks such as object detection and semantic segmentation. It outperforms MAE by 0.5 $AP^{box}$ under ViTDet Li et al. (2022) framework for object detection. Also, it outperforms MAE by 0.2 points under UperNet Xiao et al. (2018) for semantic segmentation.

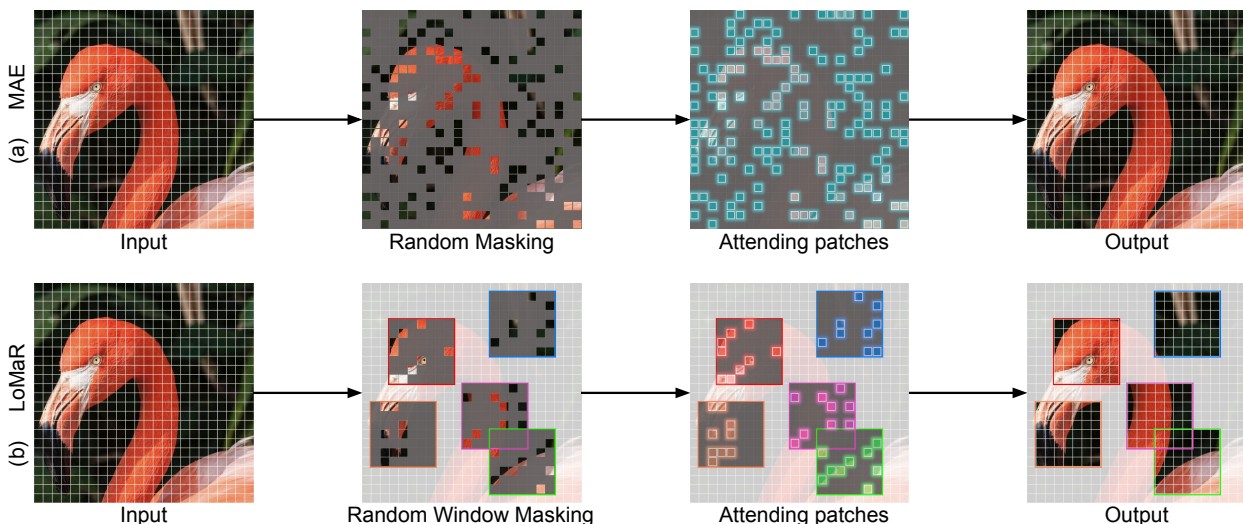

Figure 2: **Contrasting the masking and reconstruction strategy between MAE and LoMaR.** During the pretraining, MAE random mask 75% patches as masking and reconstruct them by attending to the remaining visible patches. For LoMaR, it randomly samples several small regions and masks a random subset of patches from each region, e.g. 80%. The masked patches will only attend to the visible patches inside each region for reconstruction. In contrast to MAE, LoMaR usually samples less visible patches per image.

- LoMaR is efficient and can be easily integrated into any other generative self-supervised learning approach. Equipping our local masked reconstruction learning mechanism into BEiT can improve its ImageNet-1K classification performance from 83.2 to 83.4 Top-1 accuracy, costing only 35.8% of its original pretraining time.

## 1.1 Related Work

**Self-supervised Learning For Images.** The past few years have witnessed the boom of self-supervised learning. Existing techniques can be roughly categorized as discriminative (Becker & Hinton, 1992);Pathak et al. (2017); Gidaris et al. (2018) and generative. The prominent discriminative approach, instance discrimination, distinguishes different views of the same data instance from other instances Wu et al. (2018); Oord et al. (2018); He et al. (2020); Chen et al. (2020b); Grill et al. (2020);(Chen & He, 2021). The most representative works include BYOL Grill et al. (2020) , MOCO Chen et al. (2020d; 2021); He et al. (2020) and SimCLR Chen et al. (2020b;c). VICRegLBardes et al. (2022) performs the contrasting learning in both local and global features. Other approaches such as SwAV Caron et al. (2020) and DINO Caron et al. (2021) heavily rely on multi-crop strategy for instance discrimination. The generative approach includes autoregressive prediction Han et al. (2019); Chen et al. (2020a) and autoencoders, which we discuss next.

**Autoencoders for Representation Learning.** An autoencoder, which aims to learn a representation from which the original input can be reconstructed, has been a popular choice for representation learning since the dawn of deep learning (Hinton & Salakhutdinov, 2006);Bengio et al. (2006; 2013). The autoencoding problem is inherently ill-posed due to the existence of a trivial solution: a network entirely composed of identity mappings. Hence, it is necessary to apply some form of regularization, such as sparsity Ranzato et al. (2006), input corruption Vincent et al. (2008), probability priors (Kingma & Welling, 2013);Rezende et al. (2014), or adversarial discriminators Makhzani et al. (2015).

In particular, denoising autoencoder Vincent et al. (2008), which attempts to recover the original input from a corrupted version, has received significant research attention. Variations include solving jigsaw puzzles (Noroozi & Favaro, 2016), color restoration Zhang et al. (2016); Larsson et al. (2016), spatial relation recovery Doersch et al. (2015), inpainting Pathak et al. (2016), and so on. Recently, BEiT Bao et al. (2022) proposes

to encode image patches as a dictionary using dVAE Ramesh et al. (2021) and predict the encoding of missing patches. PeCo Dong et al. (2021) further improves BEiT by enforcing perceptual similarity from dVAE. MAE He et al. (2021) reconstructs directly the missing pixels. CiM Fang et al. (2022) replaces image patches with plausible alternatives and learns to recover the original and predict which patches are replaced. Data2vec Baevski et al. (2022) performs self-supervised learning across multi-modalities. MultiMAE Bachmann et al. (2022) shows that pretraining on multi-modalities can be more training-efficient than on single-modality.

## 1.2 Approach

LoMaR relies on a stack of Transformer Vaswani et al. (2017) blocks to pretrain a large amount of unlabeled images by recovering the missing patches from corrupted images similar to MAE He et al. (2021), but LoMaR differentiates from MAE in several key places. In this section, we first revisit the MAE model and then describe the differences between LoMaR and MAE.

## 1.3 Background: Masked Autoencoder

The Masked Autoencoder (MAE) model He et al. (2021), employs an asymmetric encoder-decoder architecture. The encoder takes in a subset of patches from an image and outputs latent representations for the patches. From those, the decoder reconstructs the missing patches. For an input image with resolution $h \times w$, MAE first divides it into a sequence of non-overlapping patches. Then, MAE randomly masks out a large proportion (e.g., 75%) of image patches, see the upper side of Fig. 2. The positional encodings are added to each patch to indicate their spatial location. MAE first encodes the remaining patches into the latent representation space and then feeds the latent representations together with placeholders for the masked patches into the decoder, which carries out the reconstruction. For each reconstructed image, MAE uses the mean squared error (MSE) with the original image in the pixel space as the loss function.

| Method | Resolution | ImageNet 1K | | | Inaturalist | | |
| --- | --- | --- | --- | --- | --- | --- | --- |
| | | Time (h) ↓ | Top-1 Acc | Speed-up | Time (h) ↓ | Top-1 Acc | Speed-up |
| BEiT Bao et al. (2022) | 384 | ~408 | 83.7 | 1.0× | ~93 | 81.1 | 1.0× |
| MAE | 384 | ~203 | 84.3 | 2× | ~46 | 81.2 | 2.0× |
| LoMaR (ours) | 384 | **~81** | **84.5** | **5.0×** | **~22** | **81.4** | **4.2×** |
| BEiT Bao et al. (2022) | 448 | ~595 | 84.1 | 1.0× | ~121 | 82.3 | 1.0× |
| MAE | 448 | ~345 | 84.5 | 1.7× | ~70 | **82.4** | 1.7× |
| LoMaR (ours) | 448 | **~113** | **84.7** | **5.3×** | **~32** | 82.3 | **3.8×** |

Table 1: High-resolution image pretraining and classification results on ImageNet-1K dataset Deng et al. (2009) and Inaturalist Van Horn et al. (2018). The pretraining time are all computed on 4 NVIDIA 80GB A100 GPUs. We take BEiT Bao et al. (2022) as the comparison baseline when computing the speed-up for MAE He et al. (2021) and LoMaR. Our LoMaR can always achieve comparable or higher performance with at least 3.8× speed-up than BEiT and 2.2× speed-up than MAE on both datasets.

## 1.4 Local Masked Reconstruction (LoMaR)

We describe LoMaR by contrasting it with MAE from the following perspectives.

**Local vs. Global Masked Reconstruction.** MAE reconstructs each missing patch with patches sampled from the entire image. However, as indicated by Fig 1, usually only the patches in the proximity of the target patch contribute significantly to the reconstruction, suggesting that local information is sufficient for reconstruction. Therefore, we perform the random window masking and reconstruction on patches within a small region, shown in the bottom side of Fig. 2. Specifically, we perform the random window masking by sampling several small regions from the image and restrict the masked patches to only attend to its local surrounding visible patches, as we highlighted. Experiments find that a region size of 7×7 patches leads to the best trade-off between accuracy and efficiency. On the other hand, similar to convolutional networks

He et al. (2016);(Simonyan & Zisserman, 2014), LoMaR has the translation invariance property due to the usage of small windows sampled in random spatial locations each iteration.

From the complexity perspective, the local masking and reconstruction are more computationally efficient than the global masking and reconstruction of MAE due to fewer tokens for operation. Suppose that each image can be divided into $h \times w$ patches. The time complexity for computing the self-attention is $\mathcal{O}(h^2w^2)$. The complexity is quadratic to the number of patches and hard to scale up with large $hw$. However, for our local masked reconstruction, we sample $n$ windows where each contains $m \times m$ patches; Its computational complexity is $\mathcal{O}(hw + nm^4)$, which has linear time complexity if we fix $m \times m$ as a constant window size. It can reduce the computational cost significantly if $nm^4 \ll h^2w^2$. For example, for a 448x448 image, the cost of self-attention calculation is reduced from $448^2$ x $448^2$ in the case of MAE, to $4 \text{x} 7^2 \text{x} 7^2$ in the case of LoMaR when we sample 4 views of 7×7 patches.

**Architecture.** Instead of the asymmetric encoder-decoder of MAE, LoMaR only applies a simple Transformer encoder architecture. We input all the visible and masked patches under a sampled region into the encoder and reconstruct the masked patches through a simple MLP layer. Although feeding the masked patches into the encoder can be deemed a less efficient operation than MAE which only inputs masked patches into the decoder, we find that inputting the masks in the early stages can enhance the visual representation and make it more robust to mask reconstruction from the smaller regions. It might be because the encoder can convert the masked patches back to their original RGB representation after multiple encoder layers' interaction with the other visible patches. Those recovered masks in the hidden layers can implicitly contribute to the image representation. Therefore, LoMaR preserves the mask patches as the encoder input.

**Implementation.** Given an image, we first divide it into several non-overlapping patches. Each patch is linearly projected into an embedding. We randomly sample several square-shaped regions consisting of $K \times K$ patches at different spatial locations. We then zero out a fixed percentage of patches within each region. After that, we feed all the patches, including visible and masked ones, from each region to the encoder in raster order. We also apply the relative positional encoding Wu et al. (2021) into our model, which can enable the translation-invariant property for the local masked reconstruction. We convert the latent representations from the encoder output back to their original feature dimension with a simple MLP head and then compute the mean squared error with the normalized ground-truth image.

## 2 Experiments

We examine the performance of LoMaR by pretraining and finetuning on ImageNet-1K Deng et al. (2009) dataset with the following procedure. First, we perform the self-supervised pretraining on the ImageNet-1K training dataset without label information. Then, we finetune the pretrained model on ImageNet-1K with supervision from the labels. During finetuning, we feed all the image patches to the model and take the average of their features as the final representation for classification. We follow the same experimental settings as MAE He et al. (2021); detailed hyperparameters can be found in the supplementary material.

### 2.1 Experiments on High-resolution Images

We evaluate our model, MAE He et al. (2021) and BEiT Bao et al. (2022) on ImageNet Deng et al. (2009) and Inaturalist Van Horn et al. (2018) datasets. We pretrain and finetune on high-resolution images such as $384 \times 384$ and $448 \times 448$ images. For MAE, we follow their default settings during pretraining; Sample 75% patches as masks. For LoMaR, we set the number of views to 6 and 9 for resolutions of 384 and 448 on the ImageNet dataset, and sample 8 and 12 views for resolutions of 384 and 448 on the Inaturalist dataset. We pretrain all the models with 300 epochs and finetune them under the same image resolution.

We summarize the results in Table 1. The results demonstrate that LoMaR outperforms other models with substantially less pretraining time, which scales linearly with the window numbers. In contrast, the pretraining time of MAE and BEiT scales quadratically as the resolution increases. As a result, LoMaR is 2.5× faster than MAE (accuracy +0.2%) and 5.0× faster than BEiT (accuracy +0.8%) on 384×384 images, and for the resolution of 448×448, it is 3.1× faster than MAE (accuracy +0.2%) and 5.3× faster than BEiT (accuracy +0.6%). On Inaturalist, LoMaR is 2.1× faster than MAE model (accuracy +0.2%) and 4.2×

| Methods | Epochs | Res | Time (h) ↓ | Top-1 Acc | Speed-up |
|---|---|---|---|---|---|
| No Pretraining | - | 224 | - | 82.3 | - |
| DINO Caron et al. (2021) | 300 | 224 | - | 82.8 | - |
| MoCov3 Chen et al. (2021) | 600 | 224 | - | 83.2 | - |
| BEiT Bao et al. (2022) Ramesh et al. (2021) | 300 | 224 | ∼107 | 82.9 | 1.0× |
| MAE He et al. (2021) | 400 | 224 | ∼58 | 83.1 | 1.8× |
| LoMaR | 300 | 224 | **∼49** | **83.3** | 2.2× |
| LoMaR$_{8\times8}$ | 400 | 224 | ∼66 | 84.3 | 1.6× |

Table 2: Image classification results on the ImageNet-1K (IN1K) dataset Deng et al. (2009). All baselines excluding LoMaR$_{8\times8}$ adopt the ViT B/16 model Dosovitskiy et al. (2020) and are pretrained on 224×224 images. LoMaR$_{8\times8}$ applies ViT B/8 as the backbone. * denotes our reproduced results based on the officially released code and pretrained models. The pretraining time are all computed on 4 NVIDIA 80GB A100 GPUs.

faster than BEiT (accuracy +0.3%), and it can produce comparable performance with the other baselines but is 3.8× faster than BEiT and 2.2× faster than MAE.

## 2.2 Experiments on Low-resolution images

Table 2 summarizes the results of different self-supervised learning approaches. All models are pretrained self-supervisedly on ImageNet-1K Deng et al. (2009) under the 224×224 resolution and finetuned on labeled ImageNet-1K. LoMaR reaches the best result of MAE, 83.6%, after only 400 epochs of pretraining. When pretrained for 1,600 epochs, its performance further improves to 84.1%. When finetuned under the 384×384 resolution, LoMaR reaches an accuracy of 85.4%, 0.6% higher than the best baseline. Overall, LoMaR outperforms strong baselines with less pretraining time.

**Efficiency analysis.** We train LoMaR, MAE He et al. (2021) and BEiT Bao et al. (2022) baselines with different pretraining epochs [100, 200, 300, 400, 800, 1,600] on 224×224 images. We compare their pretraining time v.s top-1 accuracy in Fig. 3. We carefully tuned all models to achieve the best load balancing between the GPU and the CPU and the maximal image throughput during training. We do this by adjusting ghost batch size Hoffer et al. (2017) while keeping the total batch size constant for all models. We observe that, compared to baselines, LoMaR consistently achieves the same or higher accuracy in less pretraining time. Specifically, pretraining MAE achieves 83.6% accuracy but takes about 232 hours. LoMaR reaches the same accuracy within ∼66 hours of pretraining, which is 3.5× faster. BEiT requires 285 pretraining hours to get 83.2% accuracy. In contrast, LoMaR obtains a similar result within ∼49 hours, which translates to 5.8× time savings.

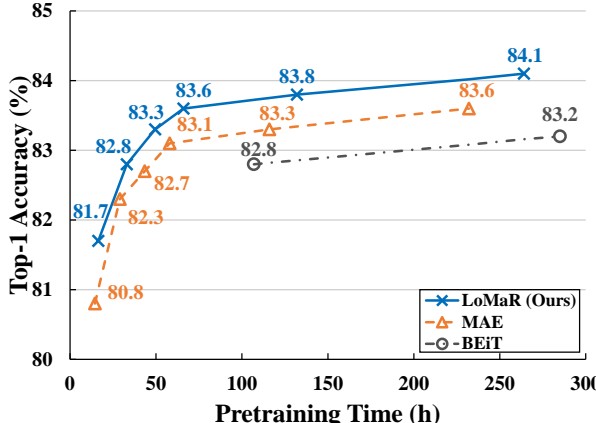

Figure 3: **Computational efficiency evaluation**: We compute their ImageNet-1K top-1 accuracy per pretraining time for low-resolution images 224×224.

**Pretraining on small patches.** We also evaluate our model on smaller patches with 8×8 pixels instead of the usual 16×16 pixels. We employ ViT B/8 Dosovitskiy et al. (2020) as a backbone in Table 2. We pretrain LoMaR with 7×7 windows (4 views per image) on 224×224 images for 400 epochs. It is worth noting that this incurs the same amount of computation time (about 66 hours) as 16×16 patches. The model accuracy after finetuning reaches to 84.3% top-1 accuracy. However, similar experiments are costly for MAE and BEiT, as smaller patches substantially increase the number of patches for operation and lead to the high

| Backbone | ViTDet*Li et al. (2022) | | ViTAE Zhang et al. (2022) | |
|----------|-----------|------------|-----------|------------|
| | $AP^{box}$ | $AP^{mask}$ | $AP^{box}$ | $AP^{mask}$ |
| MAE | 51.1* | 45.4* | 51.6 | 45.8 |
| LoMaR | 51.4 | 45.7 | 51.8 | 46.0 |
| $LoMaR_{384}$ | **51.6** | **45.9** | **52.0** | **46.2** |

Table 3: **Object detection and instance segmentation results** on MS COCO under two ViT frameworks. * denotes reproduced results with the code from Zhang et al. (2022). $LoMaR_{384}$ denotes the model pretrained on 384×384 images. Other models are pretrained on 224×224 images.

| Models | Pre-train Data | ADE20K |
|--------|---------------|--------|
| supervised | IN1K w/ labels | 47.4 |
| BEiT Bao et al. (2022) | IN1K+DALLE | 47.1 |
| MAE He et al. (2021) | IN1K | 48.1 |
| LoMaR | IN1K | 47.8 |
| $LoMaR_{384}$ | IN1K | **48.3** |

Table 4: **Semantic segmentation on ADE20K Zhou et al. (2019) (mIoU)**. All the baselines are computed under UperNet Xiao et al. (2018) framework. $LoMaR_{384}$ denotes the results under pretraining images with the resolution of 384×384. Other baselines are pretrained on 224×224 images.

cost of self-attention. In our experiments, the pretraining of MAE with their official code under smaller patches crashes due to numerical issues.

## 2.3 Object Detection and Instance Segmentation.

We finetune our model end-to-end on MS COCO Lin et al. (2014) for the object detection and instance segmentation tasks. We replace the ViT backbone with our pretrained LoMaR model in the ViTDet Li et al. (2022) and ViTAE Zhang et al. (2022) frameworks. We report object detection results in $AP^{box}$ and instance segmentation results in $AP^{mask}$.

We provide the results in Table 3. It shows the consistent improvement of LoMaR on COCO object detection benchmark. Under ViTDet, LoMaR surpasses MAE by 0.3 $AP^{box}$ and 0.3 $AP^{mask}$. When applying the LoMaR pretrained for 1,600 epochs under the 384×384 resolution, it further improves to 51.6 $AP^{box}$ and 45.9 $AP^{mask}$. In the ViTAE framework, LoMaR improves over MAE by 0.4 $AP^{box}$ and 0.4 $AP^{mask}$, respectively.

## 2.4 Semantic Segmentation

We evaluate our model on the semantic segmentation benchmark, ADE20K Zhou et al. (2019), and compare with the baselines in Table 4. We train the UperNet Xiao et al. (2018) model with our pretrained LoMaR as initialization. When applying the LoMaR pretrained under images with 384×384 resolution, it consistently outperforms MAE by 0.2 points. This shows the consistent improvement of our LoMaR over the MAE and BEiT baselines. Additionaly, it also demonstrates the usefulness of high-resolution image pretraining.

## 2.5 Integration to BEiT

Our core idea, local masked reconstruction, can be easily integrated into other generative self-supervised learning methods. To examine its effectiveness in a different paradigm, we integrate it to BEiT Bao et al. (2022). Specifically, we ran-

| Method | Time(h)↓ | Top-1 Acc | Speed-up |
|--------|----------|-----------|----------|
| BEiT | ~285 | 83.2 | 1× |
| BEiT+ window masking | ~102 | **83.4** | 2.8× |

Table 5: The results of applying our method on BEiT approach.

domly sample four 7×7 windows and feed them into the BEiT model, and pretrain for 300 epochs, while retaining all other experimental settings as the original BEiT. Results in Table 5 show that this strategy improves the accuracy from 82.8% to 83.4%, which is higher than the original BEiT and speed up the training by 2.8×.

## 2.6 Ablative Experiments

We conducted many ablation experiments to explore properties such as the window size, masking ratio, and architecture design, and share our findings in this section. We performed all the ablation experiments under 4 NVIDIA 80GB A100 GPUs with the same setting for fair comparisons, and all the experiments are obtained by pretraining under 224×224 images.

**Architecture.** Fig. 4 compares different architectures, including a simple encoder (with both visible and masked patches as input) and MAE, an asymmetric encoder-decoder architecture with local window. Initially, we sample 75% patches as the masks following the guidance of MAE. We use the absolute positional encoding (APE) for both architectures by default. We ablate these two architectures with different masked reconstruction windows, and it shows that a simple encoder can always outperform the asymmetric encoder-decoder. Moreover, the performance gap is further magnified when we decrease the window size from 14 to 7, suggesting that a simple encoder is more robust to smaller window sizes than MAE-like architecture.

**Efficiency vs. window size.** We create with multiple different window sizes such as 5×5, 7×7, 9×9, 11×11 and 14×14. One caveat is that the smaller window covers much fewer visible patches than the larger ones, which creates unfair comparisons. To encourage fairness, we assign different numbers of views for each window size, as we demonstrated in Fig. 6, thereby all conditions have similar number of visible patches in training.

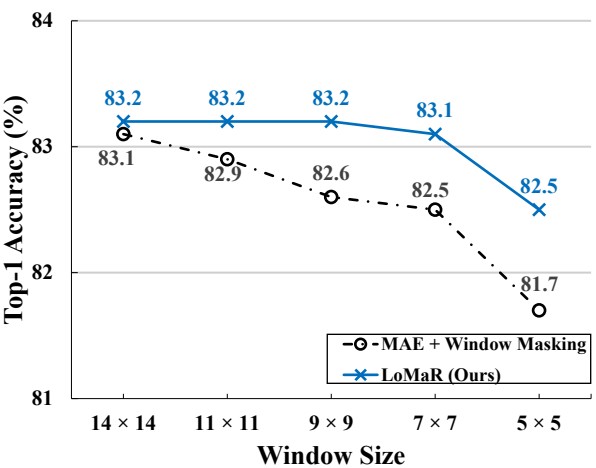

Figure 4: Comparison between LoMaR simple encoder and MAE asymmetric encoder-decoder architectures on our random window masking strategy. The window sizes vary from 14×14 to 5×5.

| Window Size | 5×5 | 7×7 | 9×9 | 11×11 | 14×14 |
|---|---|---|---|---|---|
| Views | 8 | 4 | 3 | 2 | 1 |

Table 6: **Window size ablations:** The ratio between window size and number of views per image, as utilized by LoMaR.

From the results in Fig. 4, we can observe that the performance does not change much, only 83.2 → 83.1, while decreasing the region size from 14× to 7. However, the total pretraining time decreases from 120 hours to 66 hours due to the usage of restricted attention region, meaning that pretraining on 7×7 window size can roughly 2× speed up the pretraining process with very minimal performance change. Therefore, window size 7×7 can be deemed an optimal trade-off for local masked reconstruction.

**RPE vs. APE.** Relative positional encoding (RPE) has been widely used in the previous works including BEiT Bao et al. (2022). We also employ the RPE Wu et al. (2021) in LoMaR. We observe that it can bring 0.4 top-1 accuracy gain from 83.1 to 83.5. Therefore, we set RPE as our default setting for LoMaR in our future experiments.

**Mask ratio.** We also explore the best mask ratio under the local masked reconstruction scenario (see Fig. 5). We train the previous best setting of our LoMaR on different mask ratios, ranging from 30% to 90%. The results show that too low (30%) or too high (90%) mask ratio are not optimal since they over-simplify or complicate the training task. We found that the 80% mask ratio can result in the best

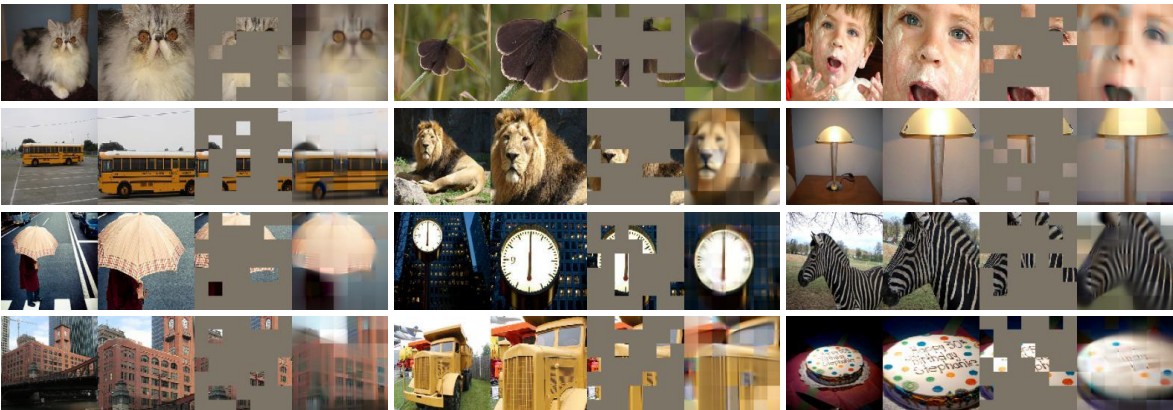

Figure 6: Example results on ImageNet (upper two rows) and COCO (lower two rows) validation images. We mask 80% patches out and reconstruct them with our pretrained model. For each image reconstruction figure, we split them into 4 parts: 1) the left-most is the original image. 2) the second-left is the sampled window (7×7 patches). 3) The second-right is the masked image. 4) The right-most is our reconstructed image.

performance, differentiating from the 60% mask ratio observed in MAE for best finetuning performance. With this motivation, we employ the 80% mask ratio in our experiments.

## 2.7 Visualization of Reconstructed Images

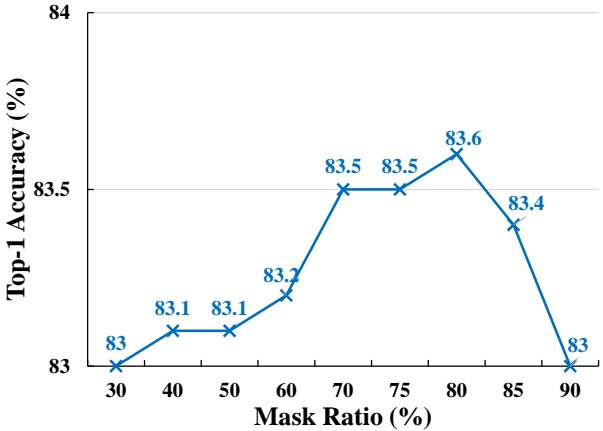

Figure 5: **Mask ratio ablations**: It compares the LoMaR under different mask ratios from 30% to 90%

We qualitatively show the reconstruction performance of our pretrained model in Fig. 6. We randomly sample several images from ImageNet-1K Deng et al. (2009) and MS COCOLin et al. (2014). After that, we sample a region containing 7×7 patches in every image and zero out 80% patches in the window for reconstruction. It can be seen that LoMaR is capable of generating plausible images, which also confirms our initial conjecture that the missing patches can be recovered from the local surrounding patches alone.

We show the reconstruction performance under different masking ratios in Fig. 7. For each image, we sample a window and randomly mask 60%, 70%, 80%, and 90% patches for reconstruction. We found that LoMaR can plausibly recover the corrupted image in various masking scales; It can still successfully reconstruct the images even with only 5 visible

patches as the clues (90% masking). This indicates that LoMaR has learned high-capacity models and can infer complex reconstructions.

## 3 Discussion and Limitation

Self-supervised learning (SSL) can benefit from training with massive amount of unlabeled data, which has brought many promising results Kenton & Toutanova (2019); Radford et al. (2018; 2019); Brown et al. (2020); He et al. (2021); Bao et al. (2022); Chen et al. (2021). However, their high computational demands remain a significant concern under large-scale pretraining. In our study, we observe that the local masked reconstruction (LoMaR) for generative SSL is more efficient than the global version used by the influential

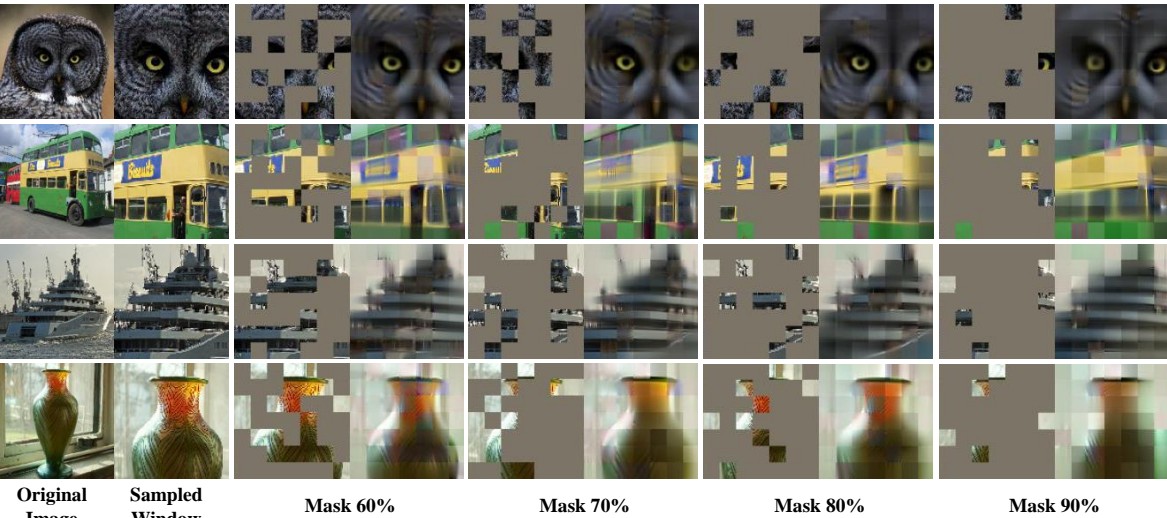

**Original Image** **Sampled Window** **Mask 60%** **Mask 70%** **Mask 80%** **Mask 90%**

Figure 7: Reconstruction examples of sampled ImageNet (upper two rows) and COCO (lower two rows) validation images under different masking ratios ranging from 60% to 90%.

works of MAE He et al. (2021) and BEiT Bao et al. (2022). LoMaR demonstrates good generalization in image classification, instance segmentation and object detection; it can be easily incorporated into both MAE and BEiT. LoMaR holds the promise to scale up SSL to even bigger datasets and higher resolution Radford et al. (2021); Sun et al. (2017) as well as more computation-intensive datasets such as videos Miech et al. (2019).

Another advantage of LoMaR lies in efficiency gain when the number of image patches increases, such as for the high-resolution images such as 384×384 and 448×448 or even larger. The primary reason is that LoMaR restricts the self-attention within a small region, and its computational complexity grows linearly with the number of sampled regions per image. This characteristic enables efficient pretraining under high image resolution, which would be prohibitively expensive for other SSL methods. It can benefit many vision tasks such as object detection or instance segmentation, which require dense prediction at the pixel level.

Despite the high pretraininig efficiency gain of LoMaR over other baselines for high-resolution images, one limitation is that LoMaR underperforms in linear probing (see results in Supplementary), which is mainly due to two reasons: 1) There is a discrepancy between training and inference. During pretraining, we feed only a small region of patches, along with masked tokens, to the network. During linear probing, the input contains all image patches without masked tokens, resulting in a shift of input distribution and damages linear probing performance. 2) LoMaR applies a much shallower decoder than MAE. A deep decoder improves linear probing performance because that the last few layers in an autoencoder are specialized for reconstruction and not very helpful for recognition; MAE removes these layers during linear probing. However, as shown in Table 2, this limitation can be easily mitigated by fine-tuning the entire model. We hope the idea of local masked reconstruction idea, as pioneered by LoMaR, can lead to further research on efficient self-supervised learning.

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
