# OpenReview forum: "Local Masked Reconstruction for Efficient Self-Supervised Learning on High-resolution Images"
_TMLR — Rejected by TMLR_

### Review · Reviewer_sgvA · 2023-11-07

**Summary Of Contributions:**

Self-supervised pre-training approaches have demonstrated superior performance in various downstream tasks. One of the predominant pre-training approaches involves using generative models such as MAE [1] or BEiT [2] that reconstructs image given a masked input. A key bottleneck in the adoption of such methods is, however, their computational complexity and enormous compute requirements. Based on the insight that generative approaches rely primarily on a local neighborhood for reconstructing masked image patches, the authors proposed a local masked reconstruction based (LoMaR) pre-training. The complexity of MAE scales quadratically with the image's resolution, whereas the complexity of the proposed method LoMaR scales linearly with the resolution. The authors empirically demonstrated that LoMaR, while being significantly faster than MAE and BEiT, also generalizes better on downstream tasks such as object detection and semantic segmentation. Additionally, the local masked reconstruction, when coupled with existing methods such as BEiT, also results in performance improvement and training efficiency.

[1] He, Kaiming, Xinlei Chen, Saining Xie, Yanghao Li, Piotr Dollár, and Ross Girshick. "Masked autoencoders are scalable vision learners." In Proceedings of the IEEE/CVF conference on computer vision and pattern recognition, pp. 16000-16009. 2022.

[2] Bao, Hangbo, Li Dong, Songhao Piao, and Furu Wei. "BEiT: BERT Pre-Training of Image Transformers." In International Conference on Learning Representations. 2021.

**Audience:**

Yes

**Claims And Evidence:**

Yes

**Requested Changes:**

- In Table 2, since the difference in performance between MAE and the proposed method is marginal, it would be helpful to include the variance across multiple runs.
- In the caption of Table 2, it is mentioned that reproduced results are indicated by ‘*’. However, there are none in the table. So, it is unclear if the authors want to say that none of the methods are reproduced.
- In Section 2.6 “Efficiency vs window size”, the line reads “… we assign different numbers of views for each window size, as demonstrated in Fig. 6, …”. However, I believe it should have been Table 6 instead.

**Strengths And Weaknesses:**

Strengths:

- Although MAEs pre-training has demonstrated promising results, they are challenging to train as they require significant computing power and time. In this context, the current paper proposes a method based on empirical evidence that significantly reduces the training time and, at the same time, generalizes better to complex downstream tasks such as object detection and semantic segmentation. So, the direction of the paper is relevant and timely.
- The authors further demonstrated that LoMaR can be readily coupled with existing SOTA methods, such as BEiT, to improve performance and efficiency.
- The effectiveness of LoMaR is empirically demonstrated, and ablation studies validate the design choices.
- Overall, the paper follows a logical structure which is easy to follow.

Weaknesses:
- In Section 1.4 “Local vs Global Masked Reconstruction”, the computational complexity of LoMaR is mentioned as $O(hw+nm^4)$. However, based on my understanding and the following example, it is expected it to be $O(nm^4)$. Can you elaborate on this?

- For the discussions in Sections 2.3 and 2.4, the authors mention improvement in the performance when using LoMaR pre-trained on $384 \times 384$ images as opposed to using MAE pre-trained on $224 \times 224$ images. It raises a question as to whether it is fair to compare two methods trained with different image resolutions. Further, considering the same resolution ($224 \times 224$) of images in pre-training for both MAE and LoMaR, the difference in performance is minimal for object detection and segmentation, while MAE can be seen outperforming LoMaR for semantic segmentation. I think this needs further elaboration.

- In Section 2, the authors mention that during finetuning for classification, all the image patches are passed through the model, and the average of the extracted features is considered the final representation. However, it is not motivated why such an approach is being considered. Further in Section 3, the authors mentioned that the linear probing performance of LoMaR is lower than MAE. It would be interesting to understand if the design choice for linear probing has any impact on this observed performance.

---

> ### Author Response · Authors · 2023-12-17
>
> ``` 1. Computational complexity of LoMaR ```
>
> Yes, that is correct. We appreciate the observation, and in the revised version, we will make the necessary corrections to clarify the time complexity as $O(nm^4)$.
>
> ``` 2. LoMaR pre-trained on 384×384 images VS MAE pre-trained on 224×224 ```
>
> The primary claim of these findings is to demonstrate that LoMAR can achieve performance similar to MAE while requiring less time or a similar computational budget. LoMAR exhibits a slight superiority over MAE at 224 resolution, and this advantage becomes more pronounced at 384 resolution. Notably, the reported performance of LoMAR at 384, when compared to MAE, is considered comparable, as both models operate within the same computational budget (232 hours for MAE and 216 hours for LoMAR). To train MAE at 384 x 384 resolution, for equivalent epochs, would require over 500 hours.
>
> ``` 3. Finetuning approach ```
>
> * During pretraining we focus only on the window selected from the mask but for finetuning for the image classification, we need the entire image representation for downstream tasks. We do this by following the same practice as MAE. The masked prediction strategy during the pretraining is to make the model learn representation predictive of the image semantics. The finetuning stage is to make the model perform better for the downstream tasks, and it will have higher performance by inputting complete information about the image.
> * Lower performance on linear probing: As we discussed in our paper (Discussion and Limitation section), there are mainly two reasons for lower performance on linear probing 1) There is a discrepancy between training and inference. During pretraining, we feed only a small region of patches, along with masked tokens, to the network. During linear probing, the input contains all image patches without masked tokens, resulting in a shift of input distribution and damaging linear probing performance. 2) LoMaR applies a much shallower decoder than MAE. A deep decoder improves linear probing performance because the last few layers in an autoencoder are specialized for reconstruction and not very helpful for recognition; MAE removes these layers during linear probing. However, as shown in Table 2 (main paper), this limitation can be mitigated by fine-tuning the entire model.

---

> ### Author Response · Authors · 2023-12-17
> **Requested Changes:**
>
> ```1. Include the variance across multiple runs in Table 2. ```
>
> We will incorporate this feedback. However, It is important to note that the main goal is to achieve comparable performance to MAE but with reduced computational complexity.
>
> ```2. Table 2, reproduced results indicated by ‘*’ are missing.```
>
> The *  is meant for MAE. Thank you for pointing this out. We will update this in the revised version.
>
> ```3.  Table 6 instead of Fig. 6 ```
>
> Indeed, it should be Table 6. Thank you for pointing this out. We will correct this in the revised version.

---

### Review · Reviewer_SDkP · 2023-12-21

**Summary Of Contributions:**

The paper studies self-supervised learning with high-resolution images. To overcome the high computation cost in training transformers with high-resolution images, authors propose local masked reconstruction, which reconstructs image patches from small neighboring regions. Experiments are conducted on benchmark datasets and demonstrate the effectiveness and efficiency of the proposed method over MAE and BEiT.

**Audience:**

Yes

**Claims And Evidence:**

Yes

**Requested Changes:**

1. The novelty may not be enough for TMLR.
2. More self-supervised learning methods after MAE should be compared.

**Strengths And Weaknesses:**

Pros:
1. The paper is well-organized and easy to follow.
2. The proposed local reconstruction strategy seems reasonable.

Cons:
1. My first concern is about the novelty. It seems that the method does not need to consider different windows jointly since the local information is more important for reconstruction. In this case, the method is equal to cropping small images from high-resolution images for training, which is a widely used data augmentation. In this case, the method would be useful but may not be qualified enough for TMLR.
2. In the experiment part, the model is only compared with MAE and BEiT, while there are a huge amount of self-supervised learning methods after MAE. It would be good to demonstrate the superiority of the proposed method compared with the following works.
3.  In the complexity analysis part, why the computation cost is (hw+nmm) instead of (nmm)?

---

> ### Author Response · Authors · 2024-02-06
>
> `1. The novelty may not be enough for TMLR.`
>
> Our method is different from cropping small images from high-resolution images for training. Taking MAE as an example when it crops the image 448x448 into 224x224 regions, cropping small images from high-resolution images still require inputting a large number of patches as the input (196 x 0.25 = 49 patches for MAE encoder input, and 196 patches for MAE decoder input). However, LoMaR only takes (49 x 0.2 = 10 patches for LoMaR encoder input, and 49 patches for decoder input).  This difference in the amount of input patches will significantly influence the training efficiency, especially for the high-resolution images. This is one of the most important novelty for improving the training efficiency compared to previous methods.
>
> The difference between our model architecture design also renders important for the model performance. As we can see in Figure 4, directly applying window-based masking to MAE, which is equivalent to cropping the images, does not work as well as LoMaR. This model architecture is also critical to achieving good performance in contrast to previous popular architectures such as MAE.
>
> Also to our understanding, the main evaluation criteria for TMLR is that our claims to be supported by accurate, convincing evidence, which we substantiated with the provided results.

---

> ### Author Response · Authors · 2024-02-06
>
> `2. More self-supervised learning methods after MAE should be compared.`
>
> We appreciate the reviewer's suggestion for further comparing our method with a wider range of SSL techniques. Our LoMaR training strategy is a general approach and the core local masked reconstruction idea can be easily integrated into many popular self-supervised methods.  We initially focused on MAE and BEiT due to their strong recent performance and high relevance to our task.
>
> However,  we agree that broader comparisons would strengthen our work. We propose extending our comparison to include more recent methods like Masked Siamese Networks(MSN)[1], Context Autoencoder(CAE)[2], Multi-modal Multi-task Masked Autoencoders(MMAE)[3], and Semantic-MAE(SEMMAE)[4]. Below, we compare the performance of the methods after pretraining and finetuning on Imagenet-1k.
>
> | Method | Epochs | Accuracy |
> | --- | ----------- | ----------- |
> MSN| 600 | 83.4
> CAE |300  |83.6
> CAE |800  |83.8
> CAE |1600 |83.9
> MMAE|    |83.3
> SemMAE      |800 |83.3
> SemMAE(8x8) |800 |84.5
> LoMaR       |300 |83.3
> LoMaR       |400 |83.6
> LoMaR       |1600|84.1
> LoMaR(8x8)  |400 |84.3
>
> MSN: During training, MSN matches the representation of an image with randomly masked patches to the representation of the original unmasked image. This occurs solely at the encoder level, eliminating the need for a decoder and improving efficiency. MSN does not calculate the loss at pixel or token level instead it computes loss at the representation from the encoder
>
> CAE shares an architectural resemblance to MAE, except for introducing a regressor network. This network predicts the representations of masked patches, which are then utilized by the decoder to reconstruct the masked regions. As the regressor handles masked patches, the encoder can focus solely on representation learning, enhancing its effectiveness. While similar to MAE with the added regressor, CAE's computational costs remain comparable.
>
> MMAE can be viewed as an extension of MAE to encompass different modalities. A single encoder handles various modalities simultaneously, receiving unmasked patches from each modality as input. The authors employed RGB, depth, and segmentation maps. Maintaining the structure of MAE while incorporating multiple modalities slightly increases computational costs compared to MAE.
>
> SemMAE tackles the issue of image understanding lacking the clear semantic units (like words) found in language, which hinders MAE performance in vision tasks. It introduces "semantic parts" as the visual equivalent of words. These parts are identified through self-supervised learning. During training, SemMAE progresses from masking patches within individual parts to masking entire parts within an image. This strategy encourages the network to learn both intra-part patterns and inter-part relationships, resulting in superior image representations and improved performance compared to random masking.
>
> **For the computation efficiency discussion:**
>
> LoMaR utilizes the same encoder as MAE paired with a very lightweight reconstruction head. LoMaR prioritizes making high-resolution pre-training computationally efficient. Worth noting is that all aforementioned approaches, face the same challenge as standard MAE: quadratically rising computation due to self-attention. CAE, MMAE, and SEMMAE build on top of MAE architecture adding additional components. The time complexity of these approaches at best is similar to that of MAE, so for high-resolution pre-training, LoMaR is much more efficient.
>
> MSN differs from MAE as it matches the representation of an image with randomly masked patches to the representation of the original unmasked image. We calculate the time per epoch for MSN at a high resolution of 384 and 448 and compare it with LoMaR, for 384, MSN takes 0.64 hours/epoch while for LoMar it takes 0.27 hours/epoch. For 448, MSN takes 0.88 hours/epoch while for LoMar it takes 0.38 hours/epoch. This comparison also demonstrates the computational efficiency of our LoMaR model.
>
> [1] Assran, M. et al. (2022). Masked Siamese Networks for Label-Efficient Learning. In: Avidan, S., Brostow, G., Cissé, M., Farinella, G.M., Hassner, T. (eds) Computer Vision – ECCV 2022. ECCV 2022. Lecture Notes in Computer Science, vol 13691. Springer, Cham.
>
> [2] Chen, X., Ding, M., Wang, X. et al. Context Autoencoder for Self-supervised Representation Learning. Int J Comput Vis (2023). https://doi.org/10.1007/s11263-023-01852-4
>
> [3] Bachmann, R., Mizrahi, D., Atanov, A., Zamir, A. (2022). MultiMAE: Multi-modal Multi-task Masked Autoencoders. In: Avidan, S., Brostow, G., Cissé, M., Farinella, G.M., Hassner, T. (eds) Computer Vision – ECCV 2022. ECCV 2022. Lecture Notes in Computer Science, vol 13697. Springer, Cham.
>
> [4] G. Li, H. Zheng, D. Liu, C. Wang, B. Su, and C. Zheng, “SemMAE: Semantic-guided masking for learning masked autoencoders,” in Advances in Neural Information Processing Systems (A. H. Oh, A. Agarwal, D. Belgrave, and K. Cho, eds.), 2022

---

> ### Author Response · Authors · 2024-02-06
>
> `3. In the complexity analysis part, why the computation cost is (hw+nmm) instead of (nmm)?`
>
> This is a writing mistake. It should be O(nmm). Thank you for pointing this out, we will make this change in our revised version.

---

### Review · Reviewer_Q8Lq · 2024-02-10

**Summary Of Contributions:**

The paper explores techniques to reduce the computational burden of MAE during pre-training on high-resolution images, introducing a method termed LoMaR. This approach incorporates two primary modifications:

a) It adopts a window masking strategy, focusing reconstruction efforts solely on randomly sampled $n \times n$ regions within the input image, rather than processing the entire image.

b) It replaces the decoder within MAE with a lightweight MLP head.

Experimental results demonstrate that the proposed method effectively reduces the computational overhead of MAE while maintaining competitive performance on classification tasks involving high-resolution input data.

**Audience:**

Yes

**Claims And Evidence:**

No

**Requested Changes:**

Requested Changes：

1. Provide more visualization results for empirical study
2. Provide detection/segmentation results of MAE and BEiT listed in Tab.1
3. Discuss the contribution of this work in the context of SimMIM.
4. Compare with the most updated methods as discussed in the weakness.
5. Tab 2 denotes * as reproduced results, but in fact, there is no such results.

**Strengths And Weaknesses:**

Strengths:
1. The presented results show that LoMaR is more computationally efficient than MAE when pre-training on high-resolution images.

Weaknesses:
1. Unclear motivation:
LoMaR is motivated by the empirical study presented in Fig. 1. The conducted experiments only mask a single patch within each image, while MIM methods usually mask out visual patches with a high mask ratio, where the visualization results of the attention map might be different. Moreover, the pictures used for illustration mostly contain small objects, yielding biased results. It’s better to include the visualization results with larger objects, e.g., the image in Fig.2.

2. Missing important references, and in fact, the technical novelty is extremely minor.
The author did not include any paper in 2023, such as [1,2].  These two papers also explore techniques to reduce the computational demands of MAE by predicting mask regions in the latent space. However, these works were not discussed or compared in the present paper.  Additionally, this paper uses the same masking strategy as MAE but confines it to reconstructing window/patch within window/patch masking. This results in limited technical advancements (change decoder to MLPs).

[1] Xiaokang Chen, Mingyu Ding, Xiaodi Wang, Ying Xin, Shentong Mo, Yunhao Wang, Shumin Han, Ping Luo, Gang Zeng, Jingdong Wang "Context Autoencoder for Self-Supervised Representation Learning",  IJCV, 08.2023,

[2]  Mahmoud Assran, Quentin Duval, Ishan Misra, Piotr Bojanowski, Pascal Vincent, Michael Rabbat, Yann LeCun,  Nicolas Ballas, "Self-Supervised Learning from Images with a Joint-Embedding Predictive Architecture", CVPR 2023


3. Important experiments are missing:
Pre-training on high-resolution images may benefit more dense-level tasks such as detection and segmentation, which is also emphasized in this paper. However, the authors only compare LoMaR_{384} with MAE and BEiT pre-trained with $224\times 224$ images on detection and segmentation, which is less convincing. As shown in Fig.4, the window masking harms the performance of both MAE and LoMaR in classification tasks. Such a trend can still hold in dense-level tasks and is even more significant. Therefore, it’s necessary to report the dense-level performance of models presented in Tab.1, rather than only classification results.

4.  Limited impact to the community:
As mentioned in weakness 3, it’s unclear whether LoMaR can still perform competitively in dense-level tasks compared to MAE with high-resolution pre-training. Moreover, when pre-training with the normal setting, i.e. resolution of $224\times 224$, LoMaR forwards both visible and masked tokens to the encoder, resulting in high computational cost, which can be referred to in Tab.2 (0.16 hour per epoch of LoMaR VS 0.14 hour per epoch of MAE).  It limits the impact of this paper whose key contribution is to speed up MAE pre-training. It is also worth noting that the improvements compared to MAE are very limited across different settings

---

> ### Author Response · Authors · 2024-02-25
>
> ```Unclear motivation by the empirical study presented in Fig. 1.```
>
> The idea behind the visualization is to show that to construct a local area(a patch) the attention is focused on neighborhood patches and attending to all the tokens is unnecessary. This is the hypothesis that we start and show in our experiments in Tables 1, 2, 3, and 4, that using the local neighborhood of patches is enough. Another evidence is as we show in Figure 4, that using local neighbourhood information from a window of 7x7 is enough to attain the same performance as a window of 14x14, confirming that attending to the local area is already sufficient for learning good representation.

---

> > ### Author Response · Authors · 2024-02-25
> >
> > ```Limited impact to the community```
> >
> > LoMaR’s main contribution is to show competitive performance to MAE while using less computing and time for high-resolution images. While our model design is slightly slow when trained at 224 resolution, the benefits start increasing when we increase the input resolution. For 384x384 input, resolution LoMaR is 5 times faster than BeIT; at 448 resolution, it is 5.3 faster.
> >
> >
> > ```Tab 2 denotes * as reproduced results, but in fact, there is no such results.```
> >
> > The * is meant for MAE. Thank you for pointing this out. We will update this in the revised version.
> >
> > ```Discuss the contribution of this work in the context of SimMIM.```
> >
> > SimMIM[1] also uses a lightweight decoder, however, the main difference is they utilize the entire image which also makes it computationally expensive as we go high in resolution. While LoMaR’s architecture ensures the compute does not scale up quadratically due to self-attention.
> >
> > [1] Zhenda Xie, Zheng Zhang, Yue Cao, Yutong Lin, Jianmin Bao, Zhuliang Yao, Qi Dai, Han Hu (2021). SimMIM: A Simple Framework for Masked Image Modeling.

---

> > > ### Author Response · Authors · 2024-02-25
> > >
> > > ```Missing important references, and in fact, the technical novelty is extremely minor. The author did not include any paper in 2023, such as [1,2]. Additionally, this paper uses the same masking strategy as MAE but confines it to reconstructing window/patch within window/patch masking. ```
> > >
> > > - Our method is different from cropping small images from high-resolution images for training. Taking MAE as an example when it crops the image 448x448 into 224x224 regions, cropping small images from high-resolution images still require inputting a large number of patches as the input (196 x 0.25 = 49 patches for MAE encoder input, and 196 patches for MAE decoder input). However, LoMaR only takes (49 x 0.2 = 10 patches for LoMaR encoder input, and 49 patches for decoder input). This difference in the amount of input patches will significantly influence the training efficiency, especially for the high-resolution images. This is one of the most important novelty for improving the training efficiency compared to previous methods. The difference between our model architecture design also renders important for the model performance. As we can see in Figure 4, directly applying window-based masking to MAE, which is equivalent to cropping the images, does not work as well as LoMaR. This model architecture is also critical to achieving good performance in contrast to previous popular architectures such as MAE.
> > > - We compare some more recent works with LoMaR in the table below, including [1]. However, it is to be noted that we do not have a fair comparison with [2], as the main goal for  [2] is to improve performance without a finetuning approach, and they do not provide results for the finetuning setting.  We extend our comparison to include more recent methods like Masked Siamese Networks(MSN)[3], Multi-modal Multi-task Masked Autoencoders(MMAE)[4], and Semantic-MAE(SEMMAE)[5]. Below, we compare the performance of the methods after pretraining and finetuning on Imagenet-1k.
> > > | Method | Epochs | Accuracy |
> > > | --- | ----------- | ----------- |
> > > MSN| 600 | 83.4
> > > CAE |300  |83.6
> > > CAE |800  |83.8
> > > CAE |1600 |83.9
> > > MMAE|    |83.3
> > > SemMAE      |800 |83.3
> > > SemMAE(8x8) |800 |84.5
> > > LoMaR       |300 |83.3
> > > LoMaR       |400 |83.6
> > > LoMaR       |1600|84.1
> > > LoMaR(8x8)  |400 |84.3
> > >
> > >
> > > **For the computation efficiency discussion:**
> > >
> > > LoMaR utilizes the same encoder as MAE paired with a very lightweight reconstruction head. LoMaR prioritizes making high-resolution pre-training computationally efficient. Worth noting is that all aforementioned approaches, face the same challenge as standard MAE: quadratically rising computation due to self-attention. CAE, MMAE, and SEMMAE build on top of MAE architecture adding additional components. The time complexity of these approaches at best is similar to that of MAE, so for high-resolution pre-training, LoMaR is much more efficient.
> > >
> > > MSN differs from MAE as it matches the representation of an image with randomly masked patches to the representation of the original unmasked image. We calculate the time per epoch for MSN at a high resolution of 384 and 448 and compare it with LoMaR, for 384, MSN takes 0.64 hours/epoch while for LoMar it takes 0.27 hours/epoch. For 448, MSN takes 0.88 hours/epoch while for LoMar it takes 0.38 hours/epoch. This comparison also demonstrates the computational efficiency of our LoMaR model.
> > >
> > > [1] Chen, X., Ding, M., Wang, X. et al. Context Autoencoder for Self-supervised Representation Learning. Int J Comput Vis (2023). https://doi.org/10.1007/s11263-023-01852-4
> > >
> > > [2] Mahmoud Assran, Quentin Duval, Ishan Misra, Piotr Bojanowski, Pascal Vincent, Michael Rabbat, Yann LeCun, Nicolas Ballas, "Self-Supervised Learning from Images with a Joint-Embedding Predictive Architecture", CVPR 2023
> > >
> > > [3] Assran, M. et al. (2022). Masked Siamese Networks for Label-Efficient Learning. In: Avidan, S., Brostow, G., Cissé, M., Farinella, G.M., Hassner, T. (eds) Computer Vision – ECCV 2022. ECCV 2022. Lecture Notes in Computer Science, vol 13691. Springer, Cham.
> > >
> > > [4] Bachmann, R., Mizrahi, D., Atanov, A., Zamir, A. (2022). MultiMAE: Multi-modal Multi-task Masked Autoencoders. In: Avidan, S., Brostow, G., Cissé, M., Farinella, G.M., Hassner, T. (eds) Computer Vision – ECCV 2022. ECCV 2022. Lecture Notes in Computer Science, vol 13697. Springer, Cham.
> > >
> > > [5] G. Li, H. Zheng, D. Liu, C. Wang, B. Su, and C. Zheng, “SemMAE: Semantic-guided masking for learning masked autoencoders,” in Advances in Neural Information Processing Systems (A. H. Oh, A. Agarwal, D. Belgrave, and K. Cho, eds.), 2022

---

> > > > ### Author Response · Authors · 2024-02-25
> > > >
> > > > ```Important experiments are missing: Pre-training on high-resolution images may benefit more dense-level tasks such as detection and segmentation, which is also emphasized in this paper. However, the authors only compare LoMaR_{384} with MAE and BEiT pre-trained with 224×224 images on detection and segmentation, which is less convincing. As shown in Fig.4, the window masking harms the performance of both MAE and LoMaR in classification tasks. Such a trend can still hold in dense-level tasks and is even more significant.```
> > > >
> > > > - From Fig. 4, it can be seen that window masking significantly harms as MAE as compared to LoMaR. Even with the window size as small as 7x7, LoMaR achieves the same performance as MAE with double the window size(14x14). This clearly shows that LoMaR is more robust to window size. Going from window size 14x14 to 5x5, the drop is LoMaR’s performance is of 0.7% while for MAE it is 1.4%.
> > > > - Pre-training on high-resolution images may benefit more dense-level tasks and asd can be seen from Table 3 and 4, LoMaR pre-trained on high resolution images performs better than LoMaR pre-trained on 224. We agree that it can be possible that MAE/BEIT when pre-trained on high resolution images may also show better performance, but that will cost a huge amount of computation resources for both the pretraining and also finetuning stages. However, our main claim is to emphasize on the pretraining efficiency of LoMaR compared to other competitors and we take image classification as an example for the finetuning performance comparison, which can be seen from Table 1.

---

### Decision · Action_Editor_3Q57 · 2024-03-27

**Recommendation:** Reject

**Comment:**

The reviewers acknowledge the relevance of the problem tackled by this paper, the reasonable intuition behind the method, and the fact that the experiments show benefits over some baselines. However, in their final recommendations, the reviewers remained unconvinced by the comparison to recent methods in terms of accuracy, as well as the claimed efficiency of the proposed method. Ultimately, the three reviewers reached a consensus for rejection.

**Audience:**

Yes, in the sense that the general topic of the paper is of interest to the community. However, given the unconvincing evidence of the benefits of the proposed method, the paper is unlikely to have a high impact.

**Claims And Evidence:**

No. In particular, the reviewers express some concerns regarding the comparison to recent baselines in terms of accuracy, as well as in terms of efficiency. Ultimately, the reviewers remained unconvinced by the evaluation of the method's efficiency.